# Estimating the Carcinogenic Potency of Second-Hand Smoke and Aerosol from Cigarettes and Heated Tobacco Products

**DOI:** 10.3390/ijerph17228319

**Published:** 2020-11-10

**Authors:** Tomoyasu Hirano, Teiji Takei

**Affiliations:** Division of Tobacco Policy Research, Center for Cancer Control and Information Services, National Cancer Center, Tokyo 104-0045, Japan; takei-teiji@mhlw.go.jp

**Keywords:** heated tobacco products (HTPs), secondhand aerosol, policy, lifetime excess cancer risk

## Abstract

The revised Promotion Act, enforced in April 2020, allows the establishment of dedicated smoking rooms for heated tobacco products (HTPs). Since carcinogenicity assessment is unable to determine the safe level of secondhand smoke, we estimated excess lifetime cancer risk using previously reported risk factors. Assuming that nicotine inhalation is proportional to cancer potency, the lifetime cancer risk for HTP IQOS is expected to be below 10^−5^ (1/100,000), which is three orders of magnitude lower than that for cigarettes.

## 1. Introduction

Secondhand smoke is a significant threat to heath, and estimated to have killed 1.22 million people globally in 2017 [1]. Both policymakers and consumers are aware of the risks of tobacco smoke to bystanders. According to a social awareness survey conducted by the Japanese government’s public relations office in the summer of 2019, 78.4% of respondents (1292 of 1647 persons) felt uncomfortable being exposed to tobacco smoke from nearby smokers [2]. When asked where they had felt the most uncomfortable being exposed to tobacco smoke, most respondents (62.4%) indicated “restaurants, food courts and other establishments that mainly serve food”, followed by “on the street” (53.3%) and “taverns, bars, snacks and other establishments that mainly serve alcohol” (38.6%).

In response to the 2006 Surgeon General Report [3], the guidelines for implementation of Article 8 of the World Health Organization’s Framework Convention on Tobacco Control (WHO FCTC) indicate the need for total elimination of tobacco smoke in a space or environment to obtain a 100% smoke-free environment [4]. Furthermore, all indoor public places should be smoke-free. As a result, many countries have prohibited smoking indoors, although they continue to allow it in outdoor spaces, including on terraces, balconies and streets. Such measures do not protect against tobacco smoke exposure on the street and other outdoor areas. If regulations for smoke prevention measures were extended to also include outdoors spaces, smokers would be forced to smoke only in designated areas both indoors and outdoors.

A survey suggests that the majority of the Japanese public desire smoke-free environments both indoors and outdoors [2]. In the aforementioned poll, when asked what they would like the government to do to protect them from exposure to tobacco smoke, 72.6% of respondents answered “promoting the separation of smoking by setting up indoor designated smoking rooms and outdoor smoking areas”. Further, 60.6% answered “promoting a smoke-free policy in restaurants”, 57.9% answered “promoting a smoke-free policy in schools and government facilities”, 57.6% answered “outdoor measures such as those for streets and patronage”, and 56.4% answered “strengthening measures to prevent tobacco smoke from flowing out of indoor smoking rooms and outdoor smoking areas” [1]. For this reason, the revised Health Promotion Act, enforced in April 2020, allows the establishment of designated smoking rooms with very strict technical requirements and dedicated smoking rooms for heated cigarette products (HTPs). A duty of care will also be imposed for outdoor areas, and municipalities will promote the installation of smoking areas [5].

The reason for the FCTC guidelines and the 2006 Surgeon General Report’s request for a complete indoor smoking ban is the notion that there is no threshold value for toxicity from secondhand smoke [3,4]. In other words, there is no known safe level of secondhand smoke that can be determined using such measures as carcinogenicity assessment. However, one method of risk assessment adopts “the amount causing carcinogenesis at a probability of 1/100,000” as the Virtually Safe Dose (VSD). This method can be used in place of the No Observed Adverse Effect Level (NOAEL) or Tolerable Daily Intake (TDI) methods when a threshold is indeed thought to be present [6].

Therefore, we estimated excess cancer risk in bystanders exposed to cigarette smoke or HTP aerosol under usual indoor conditions using previously reported risk factors.

## 2. Methods

### 2.1. Ratios of Risk Factors and Nicotine Concentrations

Philip Morris International (PMI) studied and reported the impact of IQOS on indoor air quality in an environmentally controlled room using ventilation conditions recommended for simulating “Office”, “Residential”, and “Hospitality” environments [7]. We sought to validate the results obtained by this global tobacco company using an independent study to more appropriately evaluate the effect in real Japanese restaurant and bar environments and to compare the results against three different HTPs sold in Japan, not just IQOS, under the same conditions. Thus, the Japanese Ministry of Health, Labour and Welfare conducted a pilot exposure assessment with some risk characterization of nicotine and particulate matter (PM) from HTPs under two different conditions [8]. Using the nicotine concentration results obtained in this study, we compared the unit risk and average concentration of International Agency for Research on Cancer (IARC) type 1 and 2 carcinogens measured in tobacco smoke and HTPs and calculated the consumption risk [9]. The estimated risk ratio for HTP mainstream aerosol to tobacco smoke was 0.024.

### 2.2. Estimation Assumptions

The volume of nicotine inhaled in this scenario is represented by the nicotine concentration in mainstream smoke/aerosol and volume inhaled from 20 cigarettes/sticks per day. Because the Health Canada Intense (HCI) method is more representative of actual smoking behavior than the ISO method, we used the inhaled amount (55 mL/puff) described in the HCI method, and then, we calculated the concentration of nicotine inhaled by bystanders based on the concentration of nicotine in the room under usual indoor environmental test conditions. The human respiratory rate per day was assumed to be 20 m^3^ [6].

## 3. Results

The estimation process is summarized in Figure 1. 

We subsequently estimated that the lifetime excess cancer risk would be 1.9 × 10^−3^ for a daily consumption of 20 HTP sticks.
Lifetime excess cancer risk = (7.9 × 10^−2^) ×0.024 = 1.9 × 10^−3^.(1)

We determined that the amount of nicotine inhaled when exposed to a Mevius One cigarette and a HTP IQOS cigarette was 3.2 mg/day and 5.4× 10^−2^ mg/day, respectively. Assuming that nicotine inhalation is proportional to cancer potency, we calculated from the ratio of inhalation that the lifetime excess cancer risk would be 8.3 × 10^−3^ and 2.7 × 10^−6^, respectively.
R_cig._ = (7.9 × 10^−2^) × 3.2 (mg/day) ÷ {0.97 (mg/cig) × 20 (cig/d) × (55 mL/35 mL)} = 8.3 × 10^−3^(2)
R_IQOS_ = (1.9 × 10^−3^) × (5.4 × 10^−2^) (mg/day) ÷ {1.2 (mg/stick) × 20 (sticks/d) × (55 mL/35 mL)} = 2.7 × 10^−6^(3)

## 4. Discussion

One guide that can be used when considering whether exposure to aerosols from a designated smoking room for HTPs is tolerable under the revised Health Promotion Act [5] is the VSD of lifetime cancer risk. Under usual environmental conditions, the nicotine concentration is at a tolerable level that does not warrant any concern [8], even after considering uncertain factors, and the lifetime cancer risk is expected to be below 10^−5^ (1/100,000). This is three orders of magnitude lower than that for cigarettes smoked under the same conditions. The cancer risk associated with exposure to cigarette smoke indoors is much higher than 10^−5^ and it is unclear whether the risk outdoors is below 10^−5^. Although using a different approach to estimate lifetime risk, a study in Iran found that lifetime cancer risk due to formaldehyde exposure in an outdoor cafe that allowed cigarette smoking exceeded 10^−5^ (1/100,000) [10].

Any holistic approach should account for this unambiguous reality. In public health, scale is always important. We think it is necessary for policymakers to assess tolerance for health impact using nicotine concentrations and lifetime cancer risk. Because of the absence of a threshold, it is nearly impossible, at least in Japan, to reach a consensus for requests to make spaces 100% tobacco-free [2,5]. Further, the public finds it difficult to understand why the outdoors remain unregulated. Therefore, to establish the best policies for the public, indoor and outdoor findings should be compared and evaluated using the same methods and standards. 

The remaining challenges are the presence of a discrepancy between the predicted excess risk and excess cancer risk derived from mortality data [9]. These results should only be used as a rough estimate, given that it is not currently possible to accurately determine the VSD using these estimates. It is too early to know the long-term impact of using and being exposed to second-hand aerosols from HTPs, including clinical cancer potency [11,12]. The shape of the dose-response curve also remains unclear. Continued research is expected to refine the figures and discussions for risk assessment, and actual quantitative studies on designated smoking rooms for HTPs and outdoor smoking areas are expected to fill the current gaps in our knowledge.

Several limitations of our study warrant mention. First, detailed study is required to determine if there are significant differences in the composition of carcinogens in mainstream and environmental smoke, even though Mevius One was used as the number one selling cigarette product in this brand in Japan. Second, extrapolation of cancer risk based on nicotine concentrations from mainstream aerosol has a potential large caveat. Environmental secondhand aerosol may differ in composition from mainstream aerosol [9]. Third, although our approach was based on the chemical composition of mainstream aerosol, it would be preferable to calculate excess cancer risk based on the concentration of chemicals in environmental secondhand aerosol. However, there is also the issue of measurable limits for each substance in HTP secondhand aerosol. Since only nicotine and acetaldehydes have been detected in secondhand smoke [7], the result obtained in the present study may be an overestimate. Last, the estimated risk ratio for HTP mainstream aerosol compared with tobacco smoke, regarded as 0.024 in the present study, may vary by brand and type of HTP. Since existing studies show that mainstream HTP aerosols and cigarette smoke contain similar amounts of nicotine [13,14,15,16], the approach used in the present study is likely the most plausible at this time.

## 5. Conclusions

Exposure to aerosol from HTPs in a designated smoking room under usual conditions is estimated to be tolerable since the lifetime cancer risk is expected to be below a VSD of 10^−5^ (1/100,000), which is three orders of magnitude lower than that for cigarettes smoked under the same conditions.

## Figures and Tables

**Figure 1 ijerph-17-08319-f001:**
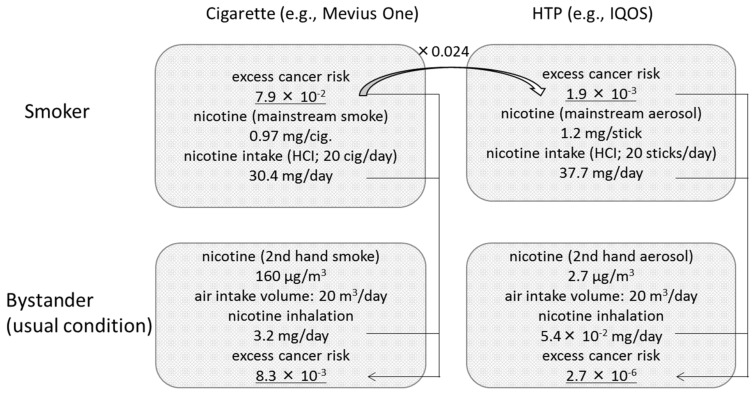
Process used to estimate excess cancer risk in bystanders exposed to cigarette smoke or heated tobacco product (HTP) aerosol under usual indoor conditions. Excess cancer risk was calculated using nicotine concentration results obtained by the Japanese Ministry of Health, Labour and Welfare, and unit risk calculated by Stephens [9].

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
