# Peer review of "Estimating the Carcinogenic Potency of Second-Hand Smoke and Aerosol from Cigarettes and Heated Tobacco Products"

_ijerph, 2020, doi:10.3390/ijerph17228319_

Round 1

Reviewer 1 Report

Peer review IJERPH-961190: Estimating the carcinogenic potency of second-hand smoke and aerosol from cigarettes and heated tobacco products, by T. Hirano and T. Takei.

The authors use a simple and straightforward method measure to estimate the excess cancer risk elicited by exposure to second hand aerosol from HTPs. Calculations are based on data published by Stephens (2017) for inhalation, and an extrapolation using measured nicotine concentrations in ‘smoker’ and ‘bystander’ conditions

General comments

The paper is very well written. The method used to estimate the excess cancer risk is simple and based on a chemical that is similar in abundance in smokers and HTP users, i.e. nicotine, and extrapolation of risk is done based on nicotine exposure. The excess cancer risk used for smokers is apparently derived from mortality data for smokers in the US in 1995 (figure mentioned in Stephens 2017), and the estimated risk ratio is derived from the same paper which is 0.024. The ratio takes the different carcinogens in the ‘mainstream smoke’ from US cigarettes and from ‘mainstream aerosol’ from the HTP product into account. This ratio may be overestimated because bystanders do not seem to be exposed to all carcinogens listed in the Stephens paper, and this according to the Mitova et al., 2016 (RTP 80(2016)91-101) paper (not cited in this article), where only nicotine and acetaldehyde could be measured in the ‘second hand’ atmosphere. The reviewer estimates that, for reasons of completeness, the overestimation should be mentioned in view of the findings in the Mitova et al paper.

Specific comments

  1. The excess cancer risk for cigarette smoke is based on US data. How representative is this for the Japanese population and de specific cigarette used (Mevius One) in the paper. Please comment on this specifically in the article. Although emissions from different cigarettes are not very different from each other, there might be a significant difference for certain of the carcinogens used in the assessment, such as acrylonitrile and 1,3-butadiene (that are the main contributing carcinogens for smoke-related cancer risk (according to Stephens).
  2. Extrapolation of cancer risk based on nicotine concentrations from mainstream smoke has a potential large caveat. It has been repeatedly demonstrated that the environmental second hand smoke has a very different composition than the mainstream smoke to which the smoker is exposed by direct inhalation. The authors should look into this to potentially correct the risk factor. If this cannot been done, this caveat should be clearly mentioned.
  3. For the risk ratio used between cigarette smoke exposure and HTP exposure (0.024), the authors use mainstream to mainstream smoke/aerosol into the equation. For smoke a remark is already mentioned under comment no. 2. For HTP emissions, the authors should also take the data from the Mitova 2016 paper into account, in which it has been demonstrated that under different conditions, only nicotine and acetaldehyde can be measured in the test room used. According to Stephens, acetaldehyde plays an important role in the cancer potency of HTP aerosol, fine, but the absence or unmeasurable levels of other carcinogens is not taken into account. Thus the best way to perform a proper risk assessment for bystander exposure should be based on data published from chemicals concentrations in the second hand smoke from cigarettes and in the environment of HTP users. In case the authors do not have sufficient data to do so, they should clearly mention this as a major limitation of the approach used.

Author Response

Our responses to the first referee's comments are as follows:

----------------------------------------------------------------------

Response for the First Referee

----------------------------------------------------------------------

General comments

The paper is very well written. The method used to estimate the excess cancer risk is simple and based on a chemical that is similar in abundance in smokers and HTP users, i.e. nicotine, and extrapolation of risk is done based on nicotine exposure. The excess cancer risk used for smokers is apparently derived from mortality data for smokers in the US in 1995 (figure mentioned in Stephens 2017), and the estimated risk ratio is derived from the same paper which is 0.024. The ratio takes the different carcinogens in the ‘mainstream smoke’ from US cigarettes and from ‘mainstream aerosol’ from the HTP product into account. This ratio may be overestimated because bystanders do not seem to be exposed to all carcinogens listed in the Stephens paper, and this according to the Mitova et al., 2016 (RTP 80(2016)91-101) paper (not cited in this article), where only nicotine and acetaldehyde could be measured in the ‘second hand’ atmosphere. The reviewer estimates that, for reasons of completeness, the overestimation should be mentioned in view of the findings in the Mitova et al paper.

As the referee has stated, Philip Morris International (PMI) studied and reported the impact of IQOS on indoor air quality in an environmentally controlled room using ventilation conditions recommended for simulating “Office”, “Residential”, and “Hospitality” environments (Mitova MI et al., 2016). Because we sought to validate the results obtained by this global tobacco company by conducting an independent study, we estimated excess cancer risk based our measurement results and the risk ratio derived by Stephens. Based on our measurements, the nicotine concentration ratio in the secondhand atmosphere to the mainstream aerosol was about 1/60, which is comparable to the 1/70 reported by Mitova et al (2016). In the case of HTPs, much of the environmental aerosol indoors likely comes from exhaled smoke because of the low production of secondhand smoke. This is based on an estimated 150 ml of anatomical dead space and approximately 30% respiration volume. Given this, we expected the composition of chemicals in aerosols in the environment and in mainstream smoke to be comparable. Studies that have conducted analysis and measurement of chemical concentrations in mainstream smoke show that in some HTPs, nicotine levels are as high as those in cigarettes, but carcinogens are generally low (Bekki K, et al. Journal of UOEH 2017;39, 201-207; Uchiyama S, et al. Chem. Res. Toxicol.2018; 31, 7, 585-593). In this paper, we estimated excess cancer risk based on the ratio of environmental concentrations indoors to mainstream aerosols to avoid overestimating because many chemical substances are below the lower limit of quantification.

Specific comments

  1. The excess cancer risk for cigarette smoke is based on US data. How representative is this for the Japanese population and de specific cigarette used (Mevius One) in the paper. Please comment on this specifically in the article. Although emissions from different cigarettes are not very different from each other, there might be a significant difference for certain of the carcinogens used in the assessment, such as acrylonitrile and 1,3-butadiene (that are the main contributing carcinogens for smoke-related cancer risk (according to Stephens).

We have added and described this in detail as a limitation in the Discussion section. Because we do not know the excess cancer risk for cigarette smoke in Japan, we have used the value stated in the study by Stephens 2017. Age-standardized mortality rates of cancer in 2018 were 85.2 and 91.0 in Japan and the US, respectively (IARC, The Global Cancer Observatory 2018), indicating a minor difference between countries. Other factors, such as historical changes in smoking prevalence, have also had a significant impact; however, because these are beyond the scope of our study, we have only discussed them briefly the limitations section. In addition, we have added details about specific cigarettes used, differences in emissions, and carcinogenic substrates as suggested [lines 117-120].

  1. Extrapolation of cancer risk based on nicotine concentrations from mainstream smoke has a potential large caveat. It has been repeatedly demonstrated that the environmental second hand smoke has a very different composition than the mainstream smoke to which the smoker is exposed by direct inhalation. The authors should look into this to potentially correct the risk factor. If this cannot been done, this caveat should be clearly mentioned.

We have added this caveat to the Discussion section, as suggested [lines 120-122].

  1. For the risk ratio used between cigarette smoke exposure and HTP exposure (0.024), the authors use mainstream to mainstream smoke/aerosol into the equation. For smoke a remark is already mentioned under comment no. 2. For HTP emissions, the authors should also take the data from the Mitova 2016 paper into account, in which it has been demonstrated that under different conditions, only nicotine and acetaldehyde can be measured in the test room used. According to Stephens, acetaldehyde plays an important role in the cancer potency of HTP aerosol, fine, but the absence or unmeasurable levels of other carcinogens is not taken into account. Thus the best way to perform a proper risk assessment for bystander exposure should be based on data published from chemicals concentrations in the second hand smoke from cigarettes and in the environment of HTP users. In case the authors do not have sufficient data to do so, they should clearly mention this as a major limitation of the approach used.

We have added the possibility of overestimation as a major limitation to the Discussion section, as suggested [line 123-130].

Reviewer 2 Report

Major comments:

While I applaud the authors for their attempt at quantifying the risk of 2nd hand HTP aerosol exposure, I am not sold on the approach. Why was nicotine chosen as the marker of interest, particularly when it is generally considered to be non-carcinogenic? Surely PM (or some actually carcinogenic component of HTP aerosol/tobacco smoke) would be more appropriate? The authors partially justify their approach by citing an unpublished pilot study which does not allow a reader to make informed decisions on the validity of the estimation.

I am also not sold on the 0.024 calculation of estimated risk ratio for HTP mainstream aerosol compared with tobacco smoke. Although results vary, there are numerous studies which show that mainstream HTP aerosols can contain similar amounts of PM and nicotine to cigarette smoke (eg Bekki K, et al. Journal of UOEH 2017; 39(3): 201-7, Schaller J-P, et al. Regulatory Toxicology and Pharmacology 2016; 81: S48-S58, Auer R, et al. JAMA Internal Medicine 2017; 177(7): 1050-2). A stronger justification of this estimated risk is required, given that the authors are stating that a bystander will be exposed to ~60 times less nicotine from HTPs compared with tobacco smoke.

Minor comments:

Abstract line 13-14 – “…excess lifetime cancer risk for HTP IQOS would be estimated 2.7 × 10-6...” doesn’t make sense to me. Please consider rewording / explaining what you mean in more detail.

Introduction line 35 – a reference needs to be cited for the sentence beginning “A survey suggests…”- even if the authors are again referring to the reference 2.

Introduction lines 39-40 – could the authors please explain what is meant by “smoke-free smoking”?

Introduction line 44 – could the authors elaborate upon “smoke prevention measures”?

Discussion line 92 – “indoor” should be “indoors”.

Discussion line 100– “remains” should be “remain”.

Author Response

Our responses to the second referee's comments are as follows:

----------------------------------------------------------------------

Response for the Second Referee

----------------------------------------------------------------------

Major comments:

While I applaud the authors for their attempt at quantifying the risk of 2nd hand HTP aerosol exposure, I am not sold on the approach. Why was nicotine chosen as the marker of interest, particularly when it is generally considered to be non-carcinogenic? Surely PM (or some actually carcinogenic component of HTP aerosol/tobacco smoke) would be more appropriate? The authors partially justify their approach by citing an unpublished pilot study which does not allow a reader to make informed decisions on the validity of the estimation.

This short communication extends the investigation and discussion provided in our original article “Exposure assessment of environmental tobacco aerosol from heated tobacco products: nicotine and PM exposures under two limited conditions” by Tomoyasu Hirano, Tokuaki Shobayashi, Teiji Takei, and Fumihiko Wakao, which was submitted at the same time as the current study for publication in Special Issue "Japan: Evaluating the Effectiveness of Tobacco Control Policies and the Use of Heated Tobacco Products" of the International Journal of Environmental Research and Public Health (ijerph-961167). That paper is currently under peer review. According to a previously published paper that examined the chemical concentrations in mainstream smoke, while some HTPs contain nicotine at similarly high concentrations to cigarettes, levels of carcinogens are generally low (Bekki K, et al. Journal of UOEH 2017;39, 201-207; Uchiyama S, et al. Chem. Res. Toxicol.2018; 31, 7, 585-593). Propylene glycol and glycerol are the most common components of PM produced by HTPs. By definition, significant amounts of “tar” are produced, but its properties and chemical composition are very different from those of cigarette smoke. We did not think PM would be a more appropriate marker because of its rapid dissipation.

We have added further descriptions to clarify our procedures to readers, as suggested [lines 58-63].

I am also not sold on the 0.024 calculation of estimated risk ratio for HTP mainstream aerosol compared with tobacco smoke. Although results vary, there are numerous studies which show that mainstream HTP aerosols can contain similar amounts of PM and nicotine to cigarette smoke (eg Bekki K, et al. Journal of UOEH 2017; 39(3): 201-7, Schaller J-P, et al. Regulatory Toxicology and Pharmacology 2016; 81: S48-S58, Auer R, et al. JAMA Internal Medicine 2017; 177(7): 1050-2). A stronger justification of this estimated risk is required, given that the authors are stating that a bystander will be exposed to ~60 times less nicotine from HTPs compared with tobacco smoke.

Because we do not know the excess cancer risk for cigarette smoke in Japan, we used the value 0.024 stated in the study by Stephens 2017. Regarding the amount of nicotine and PM in HTP aerosol, please refer to our response to the comment above.

Philip Morris International (PMI) studied and reported the impact of IQOS on indoor air quality in an environmentally controlled room using ventilation conditions recommended for simulating “Office”, “Residential”, and “Hospitality” environments (Mitova MI et al., 2016; Reg. Toxicol. Pharmacol. 80, 91-101). We conducted an independent study to validate the results obtained by this global tobacco company to estimate excess cancer risk. The nicotine concentration ratio in the secondhand atmosphere to mainstream aerosol was about 1/60 in our study, which was comparable to the 1/70 reported by Mitova et al (2016).

We have added and described this in detail as a limitation in the Discussion section [lines 128-130].

Minor comments:

Abstract line 13-14 – “…excess lifetime cancer risk for HTP IQOS would be estimated 2.7 × 10-6...” doesn’t make sense to me. Please consider rewording / explaining what you mean in more detail.

We have changed the wording to match that of the Conclusion section [line 13].

Introduction line 35 – a reference needs to be cited for the sentence beginning “A survey suggests…”- even if the authors are again referring to the reference 2.

We have cited reference No.2, as suggested [line 36].

Introduction lines 39-40 – could the authors please explain what is meant by “smoke-free smoking”?

We have revised the indicated section of text, as suggested [lines 39-40].

Introduction line 44 – could the authors elaborate upon “smoke prevention measures”?

We have revised the indicated section of text, as suggested [line 44].

Discussion line 92 – “indoor” should be “indoors”.

We have corrected the noted text [line 98].

Discussion line 100– “remains” should be “remain”.

We have corrected the noted text [line 106].

Round 2

Reviewer 2 Report

Thank you for addressing my comments.